# Demographic Drivers of Population Decline in the Endangered Korean Fir (*Abies koreana*): Insights from a Bayesian Integral Projection Model

**DOI:** 10.3390/plants14233686

**Published:** 2025-12-03

**Authors:** Jeong-Soo Park, Jaeyeon Lee, Chung-Weon Yun

**Affiliations:** 1Division of Climate Ecology Research, National Institute of Ecology, Seocheon 33657, Republic of Korea; ncc0302@nie.re.kr; 2Department of Forest Science, Kongju National University, Gongju 32588, Republic of Korea; cwyun@kongju.ac.kr

**Keywords:** demography, conservation, recruitment, survival, growth, drought

## Abstract

Understanding the demographic mechanisms underlying the decline of endangered tree species is essential for developing effective conservation strategies. This study aimed to quantify the population trajectory and its demographic drivers in the Korean fir (*Abies koreana*), a subalpine conifer endemic to South Korea and listed as endangered by the IUCN, using a Bayesian Integral Projection Model (IPM). Based on eight years of field monitoring of survival, growth, and recruitment, the Bayesian IPM estimated the population growth rate (λ_s_) and quantified its uncertainty under interannual environmental variation. The results indicated that interannual variation in drought, represented by the Standardized Precipitation–Evapotranspiration Index (SPEI), was a key driver of demographic changes. The mean population growth rate (λ = 0.983) suggests a slow decline, primarily driven by high mortality among intermediate-sized individuals, which are vital for maintaining population stability. In contrast, the growth of small to medium trees showed a weak but positive elasticity, implying that management actions targeting these size classes could benefit population persistence. Accordingly, effective conservation of *A. koreana* should focus on mitigating drought stress through reducing competition and improving soil moisture and structure.

## 1. Introduction

Population dynamics form a cornerstone of ecology, offering critical insights into species persistence and decline [1]. By integrating key processes such as survival, growth, and reproduction, demographic models capture the individual heterogeneity and life-history variation [2]. These models can be essential tools for predicting trajectories and informing conservation and management strategies [3].

Integral Projection Models (IPMs) offer a powerful approach for population dynamics by modeling continuous traits, providing more mechanistic and biologically realistic insights than traditional discrete-class matrix population models (MPMs) [1,4,5]. IPMs are particularly well-suited for slow-growing, long-lived trees compared to traditional MPMs [5,6]. A key advantage is that IPMs naturally accommodate continuous traits like size, thus avoiding the arbitrary categorization of individuals into discrete classes required by MPMs [1,7]. IPMs integrate probability density functions (for continuous variables like growth) and transition probabilities (for discrete events like survival or reproduction) into a single projection kernel, typically comprising survival–growth (P) and reproduction (F) components [4,7]. However, recent work has questioned whether IPMs are inherently more accurate or realistic than well-constructed MPMs. Using empirical data from diverse species, Doak et al. (2021) showed that key demographic outputs from IPMs and MPMs are often very similar, and that empirical sample size, rather than model type, is frequently the main determinant of model accuracy [8].

The Korean fir (*Abies koreana* E. H. Wilson) is a representative subalpine endemic conifer species in South Korea and is listed as endangered by the IUCN [9,10]. It is reported that this population has been declining since the 1980s due to a combination of global warming-induced factors, including rising temperatures, drought, and strong winds [10,11,12,13]. Predictive modeling reinforces this concern, suggesting a rapid reduction in the species’ habitat and potential extinction in many areas by the end of the century [14,15]. However, these studies have largely focused on habitat suitability and spatial distribution, without addressing the long-term population dynamics of *A. koreana*, including crucial demographic rates such as survival, growth, and recruitment.

This study aims to analyze the population dynamics of *A. koreana* using the Bayesian Integral Projection Model (IPM) framework. We pursue the following specific objectives: (1) to quantify the effects of key environmental drivers (in particular drought) on temporal variation in the population growth rate (λ_s_); (2) to project future changes in population size structure and evaluate the sensitivity of λ to changes in individual vital-rate parameters. We therefore tested the following hypotheses: (H1) the annual population growth rate (λ_s_) of *A. koreana* is positively related to SPEI, such that wetter conditions (higher SPEI) lead to higher λ_s_, and (H2) the annual population growth rate (λ_s_) of *A. koreana* is below 1 (i.e., the population is in decline). By identifying the size classes and vital rates that most strongly drive λ_s_, our IPM framework is expected to provide concrete insights for developing effective conservation strategies for *A. koreana*.

## 2. Methods

Study sites and data collection

A 1-hectare permanent plot (100 m × 100 m) was established on the west slope of Mt. Hallasan, Jeju Island (Figure 1). The bedrock type at Mt. Hallasan is composed of volcanic rock types, such as andesite and basalt [16]. The altitude at the summit of Mt. Hallasan is 1947 m and our plot is located at 1650 m. The Korean fir is primarily distributed at elevations of approximately 1500 to 1900 m above sea level. The annual average temperature in the study area is 6.4 °C. Furthermore, the annual precipitation has been nearly 5300 mm since 2010, according to data measured at the nearest Korea Meteorological Administration weather station (1 km) [17]. Based on stem density within the plot, the dominant tree species are *A. koreana* (38.8%), followed by *Prunus maximowiczii* Rupr. (24.8%), *Taxus cuspidata* Siebold & Zucc. (23.7%), and *Quercus mongolica* Fisch. ex Ledeb. (10.8%). All trees within the plot were individually tagged, and their diameter at breast height (DBH) was measured at 1.3 m above the ground. We conducted a field survey at two-year intervals from 2016 to 2024 to record demographic data, including individual growth, mortality, and the recruitment rate.

This long-term monitoring enabled us to track the dynamics of individual trees and the overall forest structure over time. We calculated recruitment and mortality rates following standard demographic approaches. Annual mortality rate (M) and annual recruitment (R) were defined as$$M=\left[{1-\left(N_{t}/N_{o}\right)}^{1/t}\right]\times 100,\;R={\left(N_{r}/N_{o}\right)}^{1/t}\times 100$$
where $N_{t}$ is the number of stems that survived from the previous survey to the current survey, $N_{r}$ is the number of newly established stems between two consecutive surveys, $N_{0}$ is the number of live stems at the previous survey, and *t* is the number of years between surveys [18].

2.Parameter estimation for Bayesian IPMs

We developed a Bayesian IPM from individual-based field data of *A. koreana*. The dataset includes observations of individual tree survival, growth, and recruitment over an eight-year period. All statistical analyses were conducted in Python, utilizing the PyMC library for Bayesian inference. To ensure the grid spans the entire range of observed data and avoids the eviction problem—where individuals move outside the modeled size range—we defined the grid boundaries. Specifically, we used 90% of the minimum observed size as the lower bound and 110% of the maximum observed size as the upper bound [3]. This buffer helps to ensure that no individuals are “evicted” from the model due to growth. The width of each bin (h) was defined as the difference between the midpoints of adjacent bins [7]. This setup allows for the accurate numerical integration of vital-rate functions across the continuous size distribution. This grid forms the basis for the IPM kernel matrices.

Vital-rate parameters (survival, growth, and recruitment) were modeled using specific functions relating the rate to individual size. The estimation of these parameters was carried out using a Bayesian framework to account for inherent uncertainty. For each parameter, we defined a weakly informative prior distribution to allow the data to drive the posterior estimates. We used Markov Chain Monte Carlo (MCMC) sampling to obtain the posterior distributions of the parameters. Survival probability was modeled using a Bayesian logistic regression. The probability of survival for an individual of size y was a function of its size, described as:$$logit\left(p\left(x\right)\right)=\;\beta_{o,\;surv}+\;\beta_{1,\;surv}\times x$$
where $\beta_{0,\;surv}$ is the survival intercept and $\beta_{1,\;surv}$ is the survival slope. Prior distribution for $\beta_{0,\;surv}$ is $\beta_{0,\;surv}\sim Normal(\mu =0,\sigma^{2}=5)$ and $\beta_{1,\;surv}$ is $\beta_{1,\;surv}\sim Normal(\mu =0,\sigma^{2}=5)$ [3,19].

The growth of an individual was modeled using a Bayesian linear regression, where the mean growth and its standard deviation were dependent on the individual’s size.$$growth\left(x\right)\sim Normal(\mu \left(x\right),\;\sigma_{growth})$$$$\mu \left(x\right)=\beta_{0,growth}+\beta_{1,growth}\times x$$
where $\beta_{0,\;growth}$ is the growth intercept, $\beta_{1,\;growth}$ is the growth slope, and $\sigma_{growth}$ is the standard deviation of growth. Prior distribution for $\beta_{0,\;surv}$ is $\beta_{0,\;surv}\sim Normal(\mu =0,\sigma^{2}=5)$, $\beta_{1,\;surv}$ is $\beta_{1,\;surv}\sim Normal(\mu =0,\sigma^{2}=5)$, and $\sigma_{growth}\sim HalfNormal(\sigma =5)$ [3,19].

Due to the absence of specific fecundity data (such as flowering records, seed productivity, and germination rates, etc.), the model relies on the observed recruitment rate [19,20]. The recruitment process was therefore modeled in two distinct components based on field observations: the size distribution of new recruits and the size-dependent fecundity rate of mature individuals [7]. At first, the size distribution of newly recruited individuals was described using a Beta distribution to account for the skewness of the size data [1,21]. The size distribution was modeled using a Beta distribution with parameters α = 1.3 and β = 2. which reflects the empirical pattern that most newly recruited individuals occur in the smallest size classes, with progressively fewer individuals at larger recruit sizes. The relative sensitivity of λ with respect to α and β were approximately +0.08 and −0.09, respectively. These specific shape parameters were selected following model performance evaluation. This distribution was employed because it accurately captured the field observation that smaller individuals had a greater probability of recruitment. Second, the per capita fecundity rate for each individual was assumed to be size-dependent. We used a non-linear function where the fecundity rate was zero for individuals below a reproductive threshold of 5.0 cm [7]. The relative sensitivity of λ with respect to reproductive threshold was approximately −0.04. Above this threshold, the fecundity rate increased proportionally to the square root of the individual’s size [22]. This relationship is mathematically expressed as:$$F\left(y,\;x\right)=R\cdot f_{rec}\left(y\right)\cdot m(x)$$$$f_{rec}\left(y\right)\;\sim \;Beta\left(\alpha ,\;\beta \right),\;\alpha =1.3,\;\beta =2$$$$m(x)\propto {\left(\max\left(x-5.0,\;0\right)\right)}^{0.5}$$
where *x* denotes the size of a parent individual in the current generation (time *t*), and *y* denotes the size of a descendant (newly recruited individual) in the subsequent generation (time *t* + 1). Finally, the fecundity kernel F(y,x) was specified as the product of the overall recruitment rate *R*, the recruitment size distribution $f_{rec}(y)$, and the size-dependent per capita fecundity function m(x). In the discretized IPM, this formulation corresponds to representing the fecundity matrix as the outer product of the recruitment size distribution vector and the per capita fecundity vector, scaled by *R*.

3.Kernel of Bayesian IPMs

The IPM kernels were constructed using the posterior distributions of the parameters estimated in the previous step. The model is represented by a kernel, *K*, which describes the transitions of individuals between size classes from one time step to the next:$$n\left(y,\;t+1\right)=\int_{L}^{U}K\left(y,\;x\right)n\left(x,\;t\right)dx=\int_{L}^{U}\left[p\left(y,x\right)+f\left(y\;,x\right)\right]n\left(x,\;t\right)dx$$
where n(y, t+1) is the DBH size distribution of the established and recruited trees at time t+1. n(x, t) is the DBH size distribution of plants at time t. L and U are the IPM in the lowest and highest size limits [4,7]. The kernel is the sum of a Survival-Growth kernel (*P*) and a Fecundity kernel (*F*). Kernel (*P*) represents transitions attributable to survival and growth, and kernel (*F*) indicates contributions of reproductive individuals given the recruit density function at the next survey. This model was implemented by applying the midpoint rule for numerical integration to obtain a high dimensional matrix (100 × 100) [7]. To ensure biological realism and avoid the “eviction” problem, we corrected the kernel by normalizing the growth matrix and then multiplying it by the size-specific survival probabilities [6]. Kernel (*F*) represents the contribution of individuals of a certain size to the next generation through reproduction.

We calculated the population growth rate (λ_s_) as the dominant eigenvalue of the mean K-kernel. To incorporate parameter uncertainty, we performed 1000 Bayesian samples via Markov Chain Monte Carlo (MCMC) [3]. Each sample generated a unique set of parameters from their posterior distributions, which were used to calculate a corresponding IPM kernel. The average of these 1000 kernels was then used as the final, representative kernel for model projections. This method effectively quantifies the uncertainty of the population growth rate (λ_s_) [23].

Model performance was evaluated using three distinct methods. First, the performance of the survival and growth sub-models was estimated by selecting the model with the lowest Akaike Information Criterion (AIC) value [1,24]. Second, during the Markov Chain Monte Carlo (MCMC) parameter estimation process, Convergence Diagnostics were employed to confirm the reliability of the parameter estimates [3]. Third, the IPM model’s predictive accuracy was evaluated by comparing its projections to both a null model and observed data. The model, trained on past data, was used to project the population structure for a future time step (e.g., year *t* + 2). A null model, assuming no change from the last observed time step (*t* + 1), was used as a baseline. The prediction errors for both models were then quantified using metrics like Root Mean Squared Error (RMSE). This approach allowed us to demonstrate that the IPM model provided a more accurate forecast of population changes than the simple null model. To evaluate the predictive performance of the Integral Projection Model (IPM), we applied 10-fold cross-validation. The dataset was partitioned into 10 subsets of equal size, and the model was iteratively trained on 9 folds and tested on the remaining fold. This process was repeated 10 times, ensuring that each data point was used exactly once for validation [18,25]. In each fold, model predictions were compared against the null model using Root Mean Square Error (RMSE).

4.Population size structure projection

To generate the projected DBH size distributions, we calculated a time-averaged kernel $\bar{K}$ by taking the mean of these four kernels and used this as a stationary approximation of the population’s size-structured dynamics. The observed DBH distribution of live stems in 2024 was used as the initial state vector $n_{0}$. Future size structures were obtained by iteratively applying the mean kernel, $n_{t+1}=\bar{K}n_{t}$, for the required number of census intervals to reach 2030 and 2040. This procedure was implemented in Python by repeatedly multiplying the mean kernel matrix by the current size-structure vector, storing the resulting projected distributions at each step [25].

5.Estimating the drought effect on population growth rate (λ)

We quantified the association between drought conditions and population growth using a Bayesian observation-error regression, which provides full posterior uncertainty and is robust for small sample sizes. For each observation time, the population growth rate was available as a point estimate with uncertainty. We modeled only response uncertainty (on λ_s_), treating the environmental predictor Standardized Precipitation Evapotranspiration Index (SPEI) as measured without error. SPEI was computed monthly water balance D=P−PET (*PET* from a monthly FAO-56/Penman–Monteith approximation) based on the meteorological factors such as precipitation, temperature, humidity, wind speed, and solar radiation, formed 24-month rolling sums of *D*, fitted a log-logistic (Fisk) distribution to those sums and mapped CDFs to standard-normal quantiles to obtain SPEI_24_ [26]. The Bayesian observation-error regression separates measurement and structural components:$$\mathrm{Observation}\;\mathrm{model}\text{:}\;{\hat{y}}_{t}\;\sim \;N(y_{t},\;\sigma_{obs,t}),$$
$$\mathrm{Structural}\;\mathrm{model}\text{:}\;y_{t}=\alpha +\beta {\tilde{x}}_{t}$$
where $y_{t}$ is the latent true log population growth rate at time *t*. $\sigma_{obs,\;t}$ is the observation standard deviation for the response at time *t*. ${\tilde{x}}_{t}$ is the standardized environmental predictor (SPEI_24_) at time *t*. Inference was performed by MCMC. We report posterior means, 95% highest density intervals (HDIs), the posterior probability of a positive effect p(β>0), and the multiplicative change in growth per +1 SD SPEI, exp(β).

6.Parameter importance analysis

To quantify the influence of uncertainty in each vital-rate parameter on the final population growth rate (λ_s_), we used the SHAP (SHapley Additive exPlanations) framework [27,28]. First, we performed a Monte Carlo simulation based on the Bayesian posterior distributions of the IPM parameters. Six core IPM parameters (survival intercept, survival slope, growth intercept, growth slope, growth SD, and recruitment rate) were randomly sampled (*N* = 1000) from their respective posterior distributions, and for each sampled parameter set we constructed the full IPM kernel and calculated the corresponding population growth rate (λ_s_). The resulting parameter samples were assembled into a predictor matrix X and the associated λ_s_ values into a response vector y, which were used to train a surrogate regression model (Gradient Boosting Regressor, scikit-learn). We then applied the SHAP framework to this surrogate model (using the shap package version 0.48.0 in Python) to compute SHAP values for each parameter across all sampled combinations. These SHAP values provide an estimate of the marginal contribution of each vital-rate parameter, and its uncertainty, to variation in λ_s_.

Additionally, we conducted a size-specific elasticity analysis of survival and growth mean to identify which DBH classes contribute most strongly to long-term population growth (λ_s_) and how changes in key vital rates at different sizes would affect λ_s_ [29,30]. Size-specific elasticities of survival were calculated by slightly increasing survival at one DBH class at a time, updating the corresponding column of the survival–growth sub-kernel, and rescaling the resulting change in λ_s_; in practice, this is equivalent to taking the partial derivative of the kernel with respect to survival at that DBH class and multiplying it by the sensitivity matrix. Similarly, we perturbed the mean of the growth distribution at each DBH class and used the same sensitivity-based approach to obtain elasticities of the growth mean.

## 3. Results

The vital-rate models and estimated parameters for the *A. koreana* demography are summarized in Table 1. Recruitment rate was exhibited substantial temporal variation, peaking at 0.034 recruits per individual during the 2018–2020 interval. This rate decreased dramatically in the subsequent two periods. Survival probability increased consistently with tree size (DBH) across all intervals, but the small slope indicates that survival changed only slightly with increasing DBH. The growth parameter remained highly stable across the four intervals, ranging narrowly from 1.004 to 1.007. This temporal consistency in the proportional growth rate stands in sharp contrast to the high inter-annual variability observed in the survival and recruitment parameters.

Figure 2 highlights a significant shift in demographic trends during survey periods. During 2018–2020, the recruitment rate was nearly double the mortality rate. This positive balance resulted in the highest population growth rate, λ = 1.057 (SD = 0.036). Indicating a substantial population expansion. In contrast, during 2022–2024, the recruitment dramatically decreased to its lowest point (0.1%), while mortality remained relatively high (1%), which led to the lowest population growth rate of the entire survey period, λ = 0.939 (SD = 0.052), signifying the sharpest population decline. Two-year cumulative SPEI values were positive only in 2018–2020, suggesting an absence of drought during this interval, in contrast to the other periods, which were characterized by negative SPEI and thus relatively drier conditions.

Table 2 summarizes the Bayesian observation-error regression linking population growth (log λ) to SPEI. The intercept (*α*) was not credibly different from zero. In contrast, the slope (*β*) was positive and credible: the 95% HDI excluded zero and the posterior probability of a positive effect was high $\left(P\left(\beta >0\right)=0.978\right)$. On the original scale, a one-SD increase in SPEI is associated with a ~3.7–3.8% increase in λ (i.e., exp(β)≈1.038).

Figure 3 illustrates the projected shifts in the DBH size-class distribution of *A. koreana* for the year 2030 and 2040. The frequency of small individuals (DBH ≤ 5 cm) increases due to sustained recruitment, high survival, and slow growth, while intermediate-sized classes (10–25 cm) decline by 2040. At the same time, the distribution across larger size classes (≥25 cm DBH) becomes more even, indicating the limited demographic transition from small to large individuals due to low survival in the intermediate classes

The SHAP summary plots showed a clear hierarchy in parameter importance, with the survival intercept and slope having the largest absolute SHAP values across all four census intervals, indicating that uncertainty in survival parameters contributed most strongly to variation in population growth rate (λ_s_). Recruitment rate had the next-largest SHAP contributions and was generally associated with positive changes in λ_s_, whereas the growth parameters (growth intercept, slope, and SD) clustered around zero, implying a comparatively minor effect on λ_s_. This pattern was consistent among census periods, emphasizing that variation in survival—rather than growth—dominates the demographic control of population growth in *A. koreana* (Figure 4).

For each year, we plotted size-specific elasticities of survival and growth mean as functions of DBH to visualize how different size classes contribute to population growth (Figure 5). The red curves show survival elasticity, which consistently peaks at intermediate DBH classes where changes in survival have the greatest effect on λ. The orange dashed curves show elasticity of the mean growth transition: growth of small–medium trees has a weak positive effect on λ, whereas growth of large trees has a negative elasticity, indicating that further increases in their size reduce population growth.

## 4. Discussions

Forests support the majority of terrestrial biodiversity and play a pivotal role in regulating the global climate, highlighting the increasing necessity of understanding forest dynamics [31,32]. Data from large permanent plots are increasingly available. In particular, Diameter at Breast Height (DBH) is the standard and most frequently collected measurement in forestry and ecological research because it allows for the assessment of tree size, growth rates, and overall health [33,34]. Despite its foundational importance, the utility of this data remains limited, particularly for applications in predictive population modeling. Specifically, traditional demographic studies of slow-growing trees confront two major obstacles when moving beyond descriptive statistics. First, traditional matrix population model demand extremely narrow size bins (e.g., 0.1–1 cm diameter) to ensure the reliability of results, as they are highly sensitive to size category width, thereby complicating practical implementation [5]. Second, the essential fecundity information—such as flowering frequency, seed productivity, and germination rates—is notoriously difficult and labor-intensive to collect over the long observational period of time [35,36]. To circumvent the methodological challenges inherent in modeling slow-growing tree populations, this study adopted the Bayesian Integral Projection Model (IPM) framework. The IPM is uniquely suited for this task because it treats DBH size as a continuous variable, using continuous regression functions across the entire size range [1]. Furthermore, we successfully demonstrated the IPM’s robustness by utilizing only the readily available field-observed recruitment rate and implementing a novel approach to estimate the Fecundity (F) kernel, decomposing the process into the size distribution of new recruits and the size-dependent reproductive output of mature individuals, which is demonstrated by improvement of the prediction performance comparing IPM model results and Null model (Figure A1).

The population growth rate (λ_s_), determined by fluctuating vital-rate components such as survival and recruitment, exhibited significant inter-annual variation. The SHAP analysis confirmed that the primary driver of this variation is the survival rate, followed by the recruitment rate. The size-specific elasticities directly supported the survival rate of intermediate-sized individuals being the most critical demographic factor for the population’s long-term persistence (Figure 5), indicating that maintaining high survival in these size classes should be a primary conservation priority. Growth of small to medium trees has a weak but positive elasticity, suggesting that proper management actions (e.g., light environment manipulation, management of competing vegetation, improvement of soil structure and moisture, and provision of protective structures) that enhance regeneration and early growth are likely to promote population growth. In contrast, growth of the largest trees has consistently negative elasticity, implying that directing resources toward further increasing the size of old individuals will yield little demographic benefit compared with supporting younger and mid-sized cohorts.

The complex nature of recruitment strongly linked with mast cycling phenomenon, which typically involves the regionally synchronous production of large seed crops at semi-regular, three-year intervals [37]. The high recruitment observed in the 2018–2020 period, in particular, may reflect the combined effect of this mast cycle and locally favorable environmental factors that enhance sapling survival. Previous studies indicate that *A. koreana* sapling survival is strongly influenced by optimal soil moisture [10]. Furthermore, the mast cycle itself is also determined by specific climate conditions, such as the difference in summer temperatures between consecutive years [38]. The high inter-annual variability and synchronous nature of seed production can act as a critical determinant of future population structure. Consequently, if climate change alters or weakens the masting pattern (e.g., by reducing the frequency of mast years or decreasing regional synchronicity), the success rate of regeneration will become increasingly unstable, posing a broad threat to the long-term population structure. The findings indicate that unstable recruitment and environmental extremes together undermine population viability, primarily through decreased survival of intermediate-sized individuals essential for long-term persistence.

Prediction of the DBH size distribution of the *A. koreana* revealed three distinctive properties compared to the current 2024 distribution (Figure 3). The frequency of small individuals (2.5 cm ≤ DBH ≤ 5 cm) is projected to gradually increase by 2030 and 2040. This accumulation likely reflects the combined effects of continuous recruitment into the small-sized class, relatively high survival rates—contrary to previous findings [39,40,41]—and very slow growth that prolongs retention within the ≤ 5 cm class. Second, the intermediate-sized individuals (10 cm ≤ DBH ≤ 25 cm) show a noticeable decrease by 2040. This decline suggests that this size range is experiencing a higher mortality and insufficient regeneration to maintain the current density. Third, the distribution becomes flatter across the large-sized individuals (DBH ≥ 25 cm). This confirms that the low survival rates in the intermediate classes fail to effectively bridge the gap between abundant small individuals and large trees. The findings emphasize that survival of intermediate-sized individuals is vital for long-term population stability. The elevated mortality rate documented in the intermediate-sized classes is likely attributable to a convergence of multiple, interacting stress factors. The subalpine environment intensifies competitive pressures through chronic climatic stress, including low temperatures, strong winds, erratic snowfall, a short growing season, and poor soil conditions that constrain physiological function. Although intermediate-sized trees grow rapidly, resource limitations and harsh climate conditions often impose chronic stress and carbon depletion. Over the past 120 years, studies in Switzerland have revealed increasing mortality among large trees, accompanied by a decline in small-tree mortality [40]. This rise in large-tree mortality may reflect the effects of stand aging and is consistent with recent findings indicating that vulnerability to drought tends to increase with tree size or age [42,43,44]. Field survey and physiological experiments consistently suggest that the decline is driven by spring drought and the species’ high sensitivity to water stress [10,45]. Our results also support the positive relation of SPEI and population growth rate (λ_s_). A one-unit change in SPEI is associated with a 3.8% change in annual population growth. Although this effect size is moderate, for a long-lived conifer such as *A. koreana* even small shifts in λ_s_ can accumulate over decades, reinforcing the conclusion that interannual variation in drought is a key driver of the observed demographic trends.

In conclusion, our analyses support H1 in showing that λ_s_ increases in wetter years, and the Bayesian observation-error regression revealed a clear positive relationship between SPEI and log λ_s_, confirming that moisture availability is a key driver of population performance in *A. koreana*. In line with H2, the annual population growth rate (λ_s_) was generally below 1 across most census intervals, indicating a slow but ongoing demographic decline under current conditions. Together, these results suggest that stabilizing local moisture regimes will be critical for slowing or reversing the decline of *A. koreana* populations. From a modelling perspective, although the relative sensitivities of λ to the fecundity parameters were small (approximately 0.04–0.09), incorporating additional empirical data on fecundity is still expected to further increase the reliability of the model. Furthermore, to strengthen the mechanistic basis of our findings, future research should incorporate more detailed eco-physiological monitoring. In particular, deploying high-resolution soil moisture probes and light sensors within stands, and linking these data to continuous measurements of tree physiological responses (e.g., photosynthesis, sap-flow), would allow us to track how individual trees experience and respond to environmental variation. Such an integrated monitoring framework would help to disentangle short-term environmental fluctuations from longer-term trends and thereby clarify the causal links between microclimatic conditions and the demographic and physiological responses of *A. koreana*. In addition, developing models that explicitly couple population dynamics with physiological responses of *A. koreana* will be essential for more clearly quantifying the impacts of climate change and for designing effective conservation strategies.

## Figures and Tables

**Figure 1 plants-14-03686-f001:**
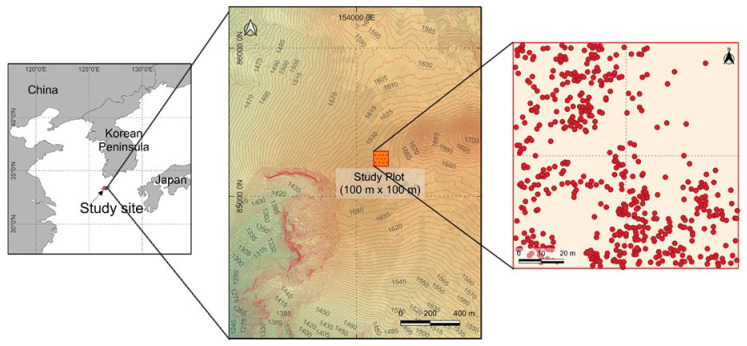
Map Showing the location of study plot (100 m × 100 m) and location of *A. koreana* (red dots) in Mt. Hallasan, Jeju Island.

**Figure 2 plants-14-03686-f002:**
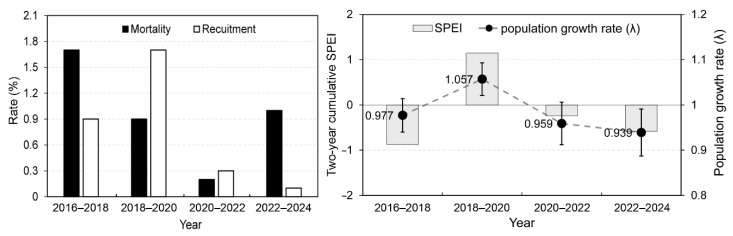
Interval variation in estimated vital-rate components (mortality and recruitment) and population growth rate (λ_s_) with two-year cumulative Standardized Precipitation Evapotranspiration Index (SPEI) over bi-annual intervals from 2016 to 2024.

**Figure 3 plants-14-03686-f003:**
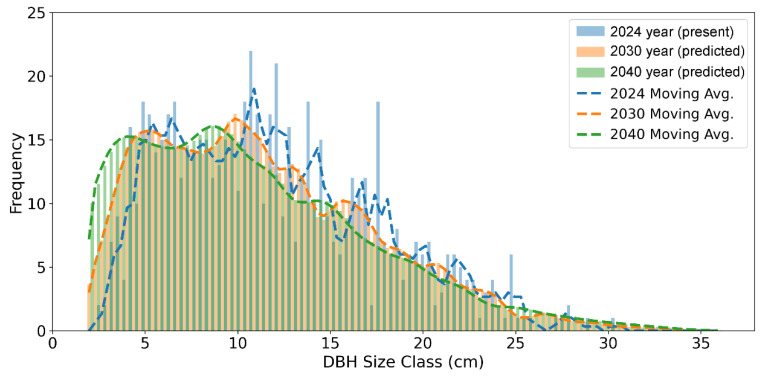
Temporal shifts in the DBH size distribution of the *A. koreana* population and comparison of the observed 2024 distribution with Integral Projection Model (IPM) predictions for 2030 and 2040.

**Figure 4 plants-14-03686-f004:**
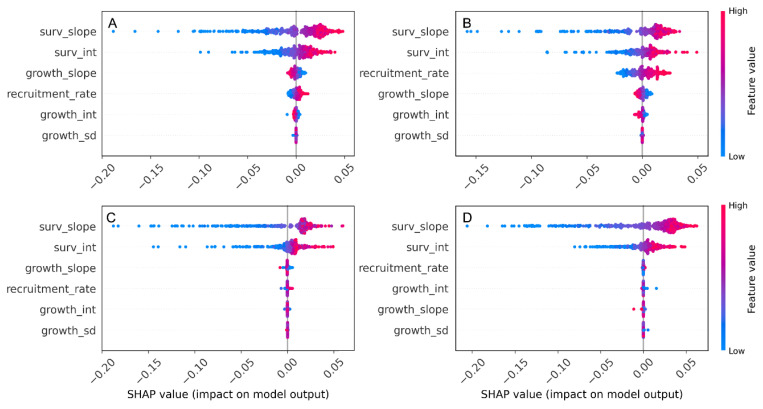
SHAP summary plots showing the relative importance and directional influence of vital-rate parameters on population growth rate (λ_s_). Each plot represents four census years: (**A**) 2016–2018, (**B**) 2018–2020, (**C**) 2020–2022, and (**D**) 2022–2024.

**Figure 5 plants-14-03686-f005:**
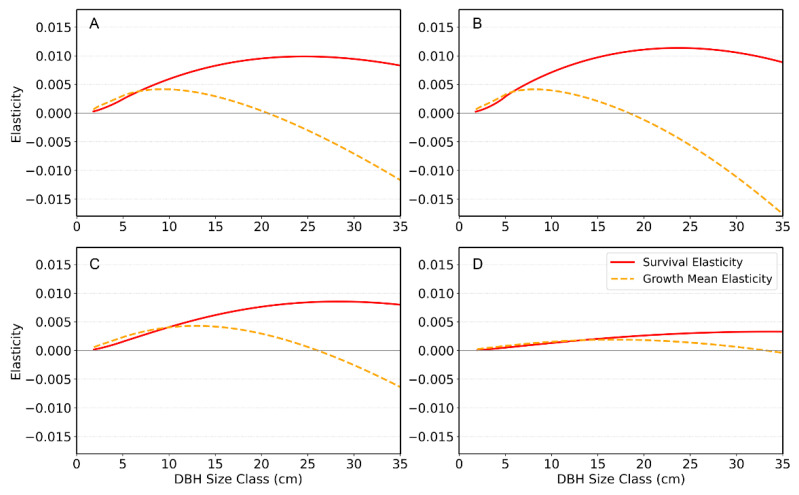
Size-specific elasticities of survival and growth for *Abies koreana* across four census years: (**A**) 2016–2018, (**B**) 2018–2020, (**C**) 2020–2022, and (**D**) 2022–2024.

**Table 1 plants-14-03686-t001:** Vital-rate models and parameter estimates for *A. koreana* demography over two-year intervals from 2016 to 2024. Model fits are evaluated using the Akaike Information Criterion (AIC). x denotes DBH at the initial census (cm) and $x^{\prime }$ is DBH at the subsequent census.

Year	Demographic Process	Vital-Rate Models	AIC
2016–2018	Survival Growth Recruitment rate	$Logit\left(p\right)=3.345+0.005x$ $x^{\prime }\sim \;N(0.27+1.007x,\sigma^{2}=0.339)$ 0.018	205.1 445.2 -
2018–2020	Survival Growth Recruitment rate	$Logit\left(p\right)=3.64+0.033x$ $x^{\prime }=N(0.303+1.005x,\sigma^{2}=0.301)$ 0.034	131.9 284.5 -
2020–2022	Survival Growth Recruitment rate	$Logit\left(p\right)=5.38+0.032x$ $x^{\prime }=N(0.272+1.003x,\sigma^{2}=0.246)$ 0.006	42.8 24.8 -
2022–2024	Survival Growth Recruitment rate	$Logit\left(p\right)=3.84+0.001x$ $x^{\prime }=N(0.323+1.004x,\sigma^{2}=0.287)$ 0.002	148.2 229.7 -

**Table 2 plants-14-03686-t002:** Summary of Bayesian observation-error regression relating log population growth rate to SPEI.

Parameter	Mean	95% HDI (Low, High)	Posterior Probability	Back-Transform
α (log-scale)	−0.011	(−0.053, 0.029)	*P* (α > 0) = 0.300	exp(α) = 0.989
β (log-scale)	0.037	(0.002, 0.073)	*P*(β > 0) = 0.978	exp(β) = 1.038

## Data Availability

The data presented in this study are available on request from the corresponding author.

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
