# Peer review of "Demographic Drivers of Population Decline in the Endangered Korean Fir (Abies koreana): Insights from a Bayesian Integral Projection Model"

_plants, 2025, doi:10.3390/plants14233686_

Round 1

Reviewer 1 Report

Comments and Suggestions for Authors

This study applies a Bayesian Integral Projection Model (IPM) to assess demographic mechanisms underlying the decline of the endangered Korean fir (Abies koreana) on Mt. Hallasan, South Korea. Using eight years of monitoring data on survival, growth, and recruitment, the authors estimate population growth rates and identify vital-rate sensitivities and size classes critical to persistence. Results indicate a gradual population decline primarily driven by elevated mortality in intermediate-sized trees, with drought stress and competition highlighted as key management concerns.

This is a valuable application of Bayesian IPMs to endangered tree species. However, several issues should be addressed, primarily in Methods. The most serious is robustness of the fecundity (F-kernel) specification. Detailed comments follow.

Lines 122-132: Fecundity is unobserved, and the F-kernel is constructed from assumptions (fixed Beta recruit-size distribution; a 5 cm reproductive threshold; fecundity ∝ size^0.5), raising concerns about robustness of conclusion. It would be helpful to provide biological justification and citations for these choices. And I would also add sensitivity analyses varying the recruit-size distribution (e.g., Beta shapes, log-normal), the reproductive threshold (e.g., 2–8 cm), and the fecundity–size function (linear, sqrt, quadratic, saturating) to support the robustness.

Line 181: The stochastic population growth rate symbol should be λs, not λ.

Line 182: “Four kernels” is unclear. Do you mean one kernel per 2-year interval? Also, I would avoid averaging kernels before eigenanalysis. Instead, compute λ for each posterior draw (build K from each draw, then take the dominant eigenvalue) and summarize the posterior of λ (median and 95% CrI)

Line 207: It is unclear for me about the meaning of the last term in the growth equation (e.g., “+ 0.339” in 2016-2018).

Line 253: Consider reporting vital rate elasticities (survival, growth, shrinkage, reproduction) or transition-type elasticities (stasis, progression, retrogression) (see Miao et al. 2025).

Line 289: Drought and competition are discussed as drivers, but no covariates enter the survival/growth models, so attribution is indirect. Where possible, incorporate stand-level structure (e.g., basal area, relative density) and climate variables into vital-rate models.

Reference:

Miao, H.-T., Salguero-Gómez, R., Shea, K., Keller, J.A., Zhang, Z., He, J.-S., et al. (2025). Differences in adult survival drive divergent demographic responses to warming on the Tibetan Plateau. Ecology, 106, e4533.

Author Response

No.

Comments

Responses

1

- Fecundity is unobserved, and the F-kernel is constructed from assumptions. It would be helpful to provide biological justification and citations for these choices.

·   Thank you for your good suggestion. We provide citations for fecundity model choices and rewrite the model equation in detail for the readers. (p.4)

2

- The stochastic population growth rate symbol should be λs, not λ.

·   As you suggested, we changed the population growth rate symbol.

3

- “Four kernels” is unclear. Do you mean one kernel per 2-year interval?

·   Thank you for your valuable suggestion. I have revised the sentence and removed the results related to long-term projections of population size. (p.4)

4

- summarize the posterior of λ (median and 95% CrI)

·   Thank you for your suggestion. However, in estimating the final value of λ, we did not adopt a fully Bayesian estimation framework. Instead, we applied Markov Chain Monte Carlo (MCMC) methods to estimate the model parameters, and constructed 1,000 K kernels to assess the uncertainty associated with λ. Based on these simulations, the standard deviation of λ has already been reported.

4

- It is unclear for me about the meaning of the last term in the growth equation

·   We appreciate your valuable suggestion. In response, we have revised the model equation and enhanced the caption of Table 1 to provide greater clarity and detail.

6

- Consider reporting vital rate elasticities or transition-type elasticities

·   We appreciate your good suggestion; we reported that size-specific elasticities of survival and growth for Abies koreana in figure 5. (p.10)

7

- Drought and competition are discussed as drivers, but no covariates enter the survival/growth models, so attribution is indirect. Where possible, incorporate stand-level structure and climate variables into vital-rate models.

·   Thank you for your valuable suggestion. We included meteorological variables in both the survival and growth models; however, these variables did not yield statistically significant effects. In contrast, we found a significant relationship between the population growth rate and the Standardized Precipitation-Evapotranspiration Index (SPEI), which integrates precipitation, temperature, humidity, wind speed, and solar radiation. This relationship has been presented and discussed in the revised Results section (Table 2).

Reviewer 2 Report

Comments and Suggestions for Authors

This submission has some strong aspects, particularly as it relates to a species of conservation concern and a modern technical approach to demographic modelling. The use of long-term data set is also noted as well as some novel applications (SHAP analysis). However, the paper fails to include alternative views of the benefits and drawbacks of IPM. In the Introduction, while touting the IPM model, the authors fail to note the paper by Doak et al. (2022) that directly compares Matrix models to IPM. Please acknowledge this in either the introduction or discussion. 

A critical comparison of integral projection and matrix projection models for demographic analysis: Reply

  Daniel F. Doak, Ellen Waddle, Ryan E. Langendorf, Allison M. Louthan, Nathalie Isabelle Chardon, Reilly Dibner, Robert K. Shriver, Cristina Linares, Maria Begoña Garcia, Sarah W. Fitzpatrick, William F. Morris, Megan L. DeMarche … See fewer authors  First published: 21 July 2022   https://doi.org/10.1002/ecy.3822  

I recommend some improvements or additional justifications in the following areas, particularly in Section 2.2

1-Introduction

It would be helpful to include either a map of the entire species distribution or at least mention it in either the introduction or methods. 

2-Methods

2.1 Study Sites and Data Collection. The meteorological data set and station are mentioned here but not specifically modelled. This seems like an underused resource. 

For DBH, list the actual height in M measured. 

For the two-year interval, why two years? Why not annually?

On page 3, the two formulas are the same but one is for recruitment and one for mortality rate. It looks like the mortality rate formula is there twice. Please check and correct. 

2.2 Parameter estimation for Bayesian IPMS (This is where most of the edits are found)

Fecundity Modeling Justification

In the fecundity modeling, why this approach (Observed Recruitment and Size Based Fecundity) was used should be supported with any field data for the 5cm DBH threshold and the square root size scaling (Page 4, 122-138).

Because density is mentioned in the discussion, why was density dependence in the recruitment function not included? Either remove or add as covariate to model. Also justify the use of a density estimator from a different species (pine vs Fir). 

Model validation section

It is recommended to provide additional clarification on the 10-fold cross validation scheme. It is presented without any details. 

As noted earlier, the meteorological stations is nearby but the model has no environmental variation included (temperature, precipitation, wind) but environmental variation is included as a major factor in the discussion. 

2.4 Population Size Projections

For the long-term projection to 2086, add in discussion of assumption of stationary demography. 

2.5 Parameter importance analysis. 

Explain the use of SHAP with more details on its application and interpretation. Most readers will not be familiar with this technique. 

Results

The results section is well organized but Figure 6 was not particularly informative. It appears to be mostly the same except for a thin slice running diagonally (but hard to read). Consider revising. 

Some minor editorial comments

Consider reducing the introduction and methods sections to remove any redundant sentences. 

Remove speculative material from the Discussion or state more clearly the proposed hypotheses.

Check terminology such as "intermediate sized"(line 251, page 8) vs "Mid-Sized" (Page 9, line 285); Be consistent in usage.

Some minor typographical and grammatical issues

For example, Page 8, line 247. "Elasticity matrix" should read "The elasticity matrix"

I recommend to add a data and code availability statement upon request. 

Author Response

No.

Comments

Responses

1

- the paper fails to include alternative views of the benefits and drawbacks of IPM.

·   We appreciate your good comments. We have included alternative views of IPMs as you recommended.

2

-  a map of the entire species distribution or at least mention it in either the introduction or methods.

·   Thank you for your opinion. We have provided the relative abundance of the dominant species in the site description section. In addition, a distribution map of Abies koreana was included to aid readers' understanding of the study area.

3

- The meteorological data set and station are mentioned here but not specifically modelled

·   Thank you for your valuable suggestion. We included meteorological variables in both the survival and growth models; however, these variables did not yield statistically significant effects. In contrast, we found a significant relationship between the population growth rate and the Standardized Precipitation-Evapotranspiration Index (SPEI), which integrates precipitation, temperature, humidity, wind speed, and solar radiation. This relationship has been presented and discussed in the revised Results section (Table 2).

4

- For DBH, list the actual height in measured.

·   We appreciate your good comments. We mentioned the actual height of DBH measured (p. 2)

5

- For the two-year interval, why two years? Why not annually?

·   To monitor changes in tree growth over time, field surveys were systematically conducted at two-year intervals. This interval was chosen to effectively capture measurable changes in individual tree DBH size in our study site.

6

- The two formulas are the same but one is for recruitment and one for mortality rate

·   Thank you for your valuable suggestion. We rewrite the recruitment and mortality formulas. (p.3)

7

- In the fecundity modeling, why this approach (Observed Recruitment and Size Based Fecundity) was used should be supported with any field data for the 5cm DBH threshold and the square root size scaling

·   Thank you for your helpful suggestion. We have added additional citations to support the choice of the fecundity model and revised the model equation in greater detail for clarity. (p.4)

8

- Because density is mentioned in the discussion, why was density dependence in the recruitment function not included? Either remove or add as covariate to model.

·   We appreciate your good comments. Although we included density information as a covariate in the model, we did not find any statistically significant association. Therefore, we excluded density-related discussion from the revised manuscript.

9

- It is recommended to provide additional clarification on the 10-fold cross validation scheme

·   We appreciate your good comments. We have provided additional clarification on the 10-fold cross-validation scheme in method and Figure A1.

10

- the meteorological stations is nearby but the model has no environmental variation included (temperature, precipitation, wind) but environmental variation is included as a major factor in the discussion.

·   Thank you for your valuable suggestion. We included meteorological variables in both the survival and growth models; however, these variables did not yield statistically significant effects. In contrast, we found a significant relationship between the population growth rate and the Standardized Precipitation-Evapotranspiration Index (SPEI), which integrates precipitation, temperature, humidity, wind speed, and solar radiation. This relationship has been presented and discussed in the revised Results section (Table 2).

11

- For the long-term projection to 2086, add in discussion of assumption of stationary demography.

·   Thank you for your helpful suggestion. We removed the long-term projection to better align with the paper’s purpose and to reduce the uncertainty associated with the projection.

12

- Explain the use of SHAP with more details on its application and interpretation

·   Thank you for your valuable suggestion. We have added explanations of SHAP in both the Methods and Discussion sections.

13

- The results section is well organized but Figure 6 was not particularly informative. It appears to be mostly the same except for a thin slice running diagonally

·   We appreciate your helpful comments. Figure 6 (the elasticity matrix) has been removed, and size-specific elasticities of survival and growth have been added to Figure 5.

14

- Consider reducing the introduction and methods sections to remove any redundant sentences.

·   Thank you for your valuable suggestion. We have carefully reviewed the manuscript and removed any redundant sentences.

15

- Remove speculative material from the Discussion or state more clearly the proposed hypotheses.

·   Thank you for your valuable suggestion. We have revised the Discussion section to focus on the proposed hypothesis.

16

- Some minor typographical and grammatical issues

·   We have carefully reviewed the manuscript and corrected the minor typographic and grammatical issues.

17

·    

Reviewer 3 Report

Comments and Suggestions for Authors

Dear authors, I reviewed the manuscript “Demographic Drivers of Population Decline in the Endangered Korean Fir (Abies koreana): Insights from a Bayesian Integral Projection Model.” In particular, I consider it a very relevant study because it generates crucial information that sheds light on the critical situation of a species considered endangered. Furthermore, the information generated can contribute to the design of conservation strategies for this species. However, there are aspects that require attention in order to improve the manuscript:
The study pursues three objectives, of which only the second is satisfactorily achieved. Regarding the first objective, no methodological evidence or concrete results were found to determine that this objective has been addressed. To validate the usefulness of the Bayesian IPM methodology, it was necessary to compare it with other methods (e.g., traditional demographic studies); it is necessary to validate the methodology using statistical parameters that allow us to understand its superiority or equality in relation to traditional methods. It is suggested that this objective be addressed separately in terms of methods, results, and discussion to improve its comprehension and contribution.
Regarding the third objective, I believe that it is not necessarily a research objective, but rather a consequence of the first two. However, the discussion section should address in depth (the presentation in the last paragraph of the discussion is deficient) the conservation strategies for Abies koreana based on the results of the study. 
Furthermore, the corresponding research hypotheses for each objective should be presented. Likewise, the conclusions should be presented, which should be formulated based on the objectives, research hypotheses, and results obtained. Other observations are presented as comments in the manuscript.

Author Response

No.

Comments

Responses

1

- Regarding the first objective, no methodological evidence or concrete results were found to determine that this objective has been addressed. To validate the usefulness of the Bayesian IPM methodology, it was necessary to compare it with other methods

·   Thank you for your valuable suggestion. We have removed the first objective and revised the manuscript to focus on the remaining two objectives. (p.1)

2

- Regarding the third objective, I believe that it is not necessarily a research objective, but rather a consequence of the first two. However, the discussion section should address in depth

·   We appreciate your helpful suggestion. We have removed the third objective and instead addressed its content in the Discussion section, based on the results of the IPMs.

3

- your comments in the pdf

·   We have carefully reviewed the manuscript and addressed the comment you mentioned in the pdf.

Round 2

Reviewer 1 Report

Comments and Suggestions for Authors

Most concerns have been resolved. Only one substantive issue remains: the biological basis of the regeneration assumptions.

Line 105: “The size distribution was modeled using a Beta distribution with α = 1.3 and β = 2.”
Line 110: “We used a non-linear function in which fecundity was zero below a reproductive threshold of 5.0 cm.”

Please (i) justify these specific values biologically (and, if possible, cite precedents), and (ii) assess their impact on the robustness of the results. A brief sensitivity analysis would suffice, for example by varying the reproductive threshold from 3 cm to 10 cm (and, ideally, exploring alternative recruit size distributions). In addition, please acknowledge these assumptions and their limitations in the final paragraph of the Discussion, clarifying how they may influence inference and management recommendations.

Author Response

Comments 1: justify these specific values (Fecundity parameters) biologically (and, if possible, cite precedents), and (ii) assess their impact on the robustness of the results.

Response 1: Thank you for your valuable comments. Following your suggestions, we assessed the impact of these parameters on the population growth rate (λ) using relative sensitivity (p.4).

Comments 2: please acknowledge these assumptions and their limitations in the final paragraph of the Discussion, clarifying how they may influence inference and management recommendations.

Response 2: Thank you for your valuable suggestions. I have described the limitations of our model in terms of the absence of field data in the Discussion section (p. 12).

Reviewer 3 Report

Comments and Suggestions for Authors

Dear authors.

The new version presented is an improvement; at least the objectives are now more in line with what was done in the research (methods, results, and discussion). However, I am still wondering why the research hypotheses and conclusions are not presented. Was the research not based on scientific hypotheses? Do the results obtained not allow valid conclusions to be drawn?

On the other hand, I am struck by what is mentioned in lines 329 to 330: “Our results also support the positive relation of drought and population growth rate (λâ‚›).”

The positive relationship between drought and population growth rate indicates that the greater the drought, the greater the population growth. Is this true? It would suggest that drought benefits population growth. Please review the wording. If the above is true, please expand on the discussion in this regard.

The above aspects need to be considered before the manuscript is published.

Author Response

Comments 1: I am still wondering why the research hypotheses and conclusions are not presented. Was the research not based on scientific hypotheses? Do the results obtained not allow valid conclusions to be drawn?

Response 2: Thank you for your comments. Following your suggestions, we have now stated the research hypotheses more clearly and explicitly presented the conclusions related to these hypotheses (p.2, p.12)

Comments 2: On the other hand, I am struck by what is mentioned in lines 329 to 330: “Our results also support the positive relation of drought and population growth rate (λâ‚›).” Please review the wording.

Response 2: Thank you for your detailed review. I made a mistake and have now revised the sentence to: “Our results also support the positive relationship between SPEI and population growth rate (λâ‚›).”
